# Monte Carlo Simulation of Stochastic Differential Equation to Study Information Geometry

**DOI:** 10.3390/e24081113

**Published:** 2022-08-12

**Authors:** Abhiram Anand Thiruthummal, Eun-jin Kim

**Affiliations:** Centre for Fluid and Complex Systems, Coventry University, Coventry CV1 5FB, UK

**Keywords:** information geometry, information length, stochastic differential equation, Langevin equation, Monte Carlo, GPU, simulation, Fokker–Planck equation, Milstein, non-linear SDE

## Abstract

Information Geometry is a useful tool to study and compare the solutions of a Stochastic Differential Equations (SDEs) for non-equilibrium systems. As an alternative method to solving the Fokker–Planck equation, we propose a new method to calculate time-dependent probability density functions (PDFs) and to study Information Geometry using Monte Carlo (MC) simulation of SDEs. Specifically, we develop a new MC SDE method to overcome the challenges in calculating a time-dependent PDF and information geometric diagnostics and to speed up simulations by utilizing GPU computing. Using MC SDE simulations, we reproduce Information Geometric scaling relations found from the Fokker–Planck method for the case of a stochastic process with linear and cubic damping terms. We showcase the advantage of MC SDE simulation over FPE solvers by calculating unequal time joint PDFs. For the linear process with a linear damping force, joint PDF is found to be a Gaussian. In contrast, for the cubic process with a cubic damping force, joint PDF exhibits a bimodal structure, even in a stationary state. This suggests a finite memory time induced by a nonlinear force. Furthermore, several power-law scalings in the characteristics of bimodal PDFs are identified and investigated.

## 1. Introduction

Stochastic Differential Equations (SDEs) (Equation (Equation 5)) are used to model various phenomena in nature, including Brownian motion, asset pricing, population dynamics, COVID-19 spread and interaction [1,2,3,4,5,6], and various other non-equilibrium processes. Due to their stochasticity, SDEs do not have an unique solution, but a distribution of solutions. A Fokker–Planck Equation (FPE) [7] is a Partial Differential Equation (PDE) that describes how the probability density of solutions of a SDE evolves with time.

Comparing solutions of different SDEs can be achieved by looking at different statistics of the solutions like mean and variance. However, when we are interested in large fluctuations and extreme events in the solutions, simple statistics might not suffice. In such cases, quantifying and comparing the time evolution of probability density functions (PDFs) of solutions will provide us with more information [8]. The time evolution of PDFs can be studied and compared through the framework of information geometry [9], wherein PDFs are considered as points on a Riemannian manifold and their time evolution can be considered as a motion on this manifold. In general, in order to have a manifold structure on the probability space in information geometry, we need to define a metric. Several different metrics can be defined on a probability space [10,11,12,13].

Different metrics have different physical and mathematical significance. For example, the Wasserstein metric (also known as the Earth mover’s distance) naturally comes up in optimal transport problems [14]; the Ruppeiner metric is based on the geometry of equilibrium thermodynamics [13]. In this work, we use a metric related to the Fisher Information [15], known as the Fisher Information metric [16,17].
(1)$$g_{jk}\left(\theta \right):=\int_{X}\frac{\partial \log p(x;\{\theta \})}{\partial \theta_{j}}\frac{\partial \log p(x;\{\theta \})}{\partial \theta_{k}}p\left(x;\left\{\theta \right\}\right)dx$$
Here, p(x;{θ}) denotes a continuous family of PDFs parametrized by parameters {θ}. This metric was used to physically represent the number of statistically distinguishable states [11,18,19]. Note that two Gaussians with same standard deviation but different means are statistically indistinguishable, if the difference in their means are much smaller than their standard deviation.

If a time-dependent PDF p(x,t) is considered as a continuous family of PDFs parameterized by a single parameter, time *t*, the scalar metric can be given by:(2)$$g\left(t\right)=\int dx\frac{1}{p(x,t)}{\left[\frac{\partial p(x,t)}{\partial t}\right]}^{2}$$
However, time in classical mechanics is a passive quantity that cannot be changed by an external control. The infinitesimal distance dL on the manifold is then given by $dL^{2}=g\left(t\right)dt^{2}$. Here, L is the Information Length defined by:(3)$$L\left(t\right):=\int_{0}^{t}dt_{1}\sqrt{\int dx\frac{1}{p(x,t_{1})}{\left[\frac{\partial p(x,t_{1})}{\partial t_{1}}\right]}^{2}}$$
The Information Length L represents the dimensionless distance, which measures the total distance traveled on the manifold, or the total number of distinguishable states a system passes through during the course of its evolution. It has previously been used in optimization problems [20]. It was also used to study dynamical systems, thermodynamics, phase transitions, memory effects, and self-organization [21,22,23,24,25,26,27,28].

The gradient of L, $\lim_{dt\to 0}dL/dt\equiv \Gamma$, then represents a velocity on this manifold
(4)$$\Gamma \left(t\right):=\sqrt{\int dx\frac{1}{p(x,t)}{\left[\frac{\partial p(x,t)}{\partial t}\right]}^{2}}$$
Note that we use the notation Γ instead of $\sqrt{g}$ to make it clear that it is a quantity defined for a time-dependent PDF. Γ represents the rate of change of statistically distinguishable states in a time-evolving PDF, and is sometimes referred to in the literature [29,30,31] as the Information Rate. Note that in information theory [32], the term “Information Rate” is used for the rate at which information is passed over the channel [33].

As for the physical significance of Γ, in a non-equilibrium thermodynamic system, Γ is related to the entropy production [30]. Γ has also been used to study causality [29] and abrupt changes in the dynamics of a system [8]. $\Gamma^{2}$ is equivalent to the (symmetric) KL divergence of infinitesimally close PDFs, as shown in Appendix E. It should also be noted that Γ(t) defined by Equation (Equation 4) has the dimensions of $t^{-1}$, and the time-integral of Γ(t) gives a dimensionless distance L in Equation (Equation 3).

Due to the lack of general mathematical techniques to solve SDEs or its associated FPE, analytical study of SDEs using Information Geometry (Γ and L) has been limited to a few special cases [24,25,34,35]. To date, numerical studies have relied on solving the associated FPE [24,34], which has the advantage of generating smooth time-dependent PDFs and information diagnostics, but has the limitations outlined in Table 1. To overcome the limitations of a FPE solver, in this work, we develop a Monte Carlo (MC) method to study time-dependent PDFs and the Information Geometry of SDEs.

The main aims of this paper are twofold. The first aim is to develop a new MC SDE simulation method and validate it by recovering the previous results obtained using the FPE method. The second is to calculate unequal time joint PDFs and investigate the effect of nonlinear forces on PDF form and various power-scaling relations. The remainder of the paper is organized as follows: Section 2.1 gives a brief introduction of the theory of MC SDE simulation. Section 2.2 develops the methods to measure Information Geometry from the simulation. Using this method, we compare a linear and a nonlinear SDE in Section 3. In Section 4, we showcase the measurement of joint PDFs of the same variable but at unequal times, which is not possible using FPE solvers. We then study a type of phase transition in the joint PDF of the nonlinear SDE and numerically verify its theoretically calculated scaling relations. Discussions are found in Section 5.

## 2. Methods

### 2.1. SDE Simulation

A typical SDE for the variable x in *d* dimensions has the following form:(5)$$d{\vec{x}}_{t}=\vec{\mu }\left({\vec{x}}_{t},t\right)\;dt+\sigma \left({\vec{x}}_{t},t\right).d{\vec{W}}_{t}$$

Here, $\vec{\mu }$ is known as the drift vector (drift coefficient in 1D), $D:=\sigma .\sigma^{T}/2$ the diffusion tensor (diffusion coefficient in 1D) and ${\vec{W}}_{t}$ is the Weiner process [37]. $d{\vec{W}}_{t}$ represents an infinitesimal random noise term, making the equation stochastic. Generalizations to SDEs can be achieved with more general noise terms and higher-order derivative terms, but are not pursued here. The associated FPE describes the time evolution of the PDF $p(\vec{x},t)$ of solutions of the SDE.
(6)$$\frac{\partial p(\vec{x},t)}{\partial t}=-\sum_{i=1}^{d}\frac{\partial }{\partial x_{i}}\left[\mu_{i}\left(\vec{x},t\right)p\left(\vec{x},t\right)\right]+\sum_{i=1}^{d}\sum_{j=1}^{d}\frac{\partial^{2}}{\partial x_{i}\partial x_{j}}\left[D_{ij}\left(\vec{x},t\right)p\left(\vec{x},t\right)\right]$$

Instead of numerically solving the FPE, in this work, we use a Monte Carlo (MC) method to estimate $p(\vec{x},t)$ by simulating a large number of instances of an SDE. Explicit numerical solution of an SDE involves choosing an initial position ${\vec{x}}_{0}$ and iteratively updating it to get the value at time *t*, ${\vec{x}}_{t}$. For the MC simulation, we start with a set of initial positions $\left\{{\vec{x}}_{0}\right\}$ sampled from the desired initial distribution and numerically solve each of them independently. We can then use the set of samples at time *t*, $\left\{{\vec{x}}_{t}\right\}$, to compute the desired statistics. We can formally write this iteration step as follows:(7)$${\vec{x}}_{t+\Delta t}=M\left({\vec{x}}_{t},\vec{\mu }\left({\vec{x}}_{t},t\right),\sigma \left({\vec{x}}_{t},t\right),\Delta {\vec{W}}_{t},\Delta t\right)$$

Here, M denotes an arbitrary update scheme. There are several methods [38] by which we can create this update. Throughout this work, we will be working with autonomous SDEs ($\vec{\mu }$ and σ do not explicitly depend on time) with diagonal noise (σ is a diagonal matrix). Therefore, we will use a simple update scheme called the Milstein scheme [39]:(8)$${\vec{x}}_{t+\Delta t}={\vec{x}}_{t}+\vec{\mu }\left({\vec{x}}_{t}\right)\Delta t+\sigma \left({\vec{x}}_{t}\right).\Delta {\vec{W}}_{t}+\frac{1}{2}\sigma \left({\vec{x}}_{t}\right).\sigma^{\prime }\left({\vec{x}}_{t}\right).\left({\left(\Delta {\vec{W}}_{t}\right)}^{2}-\Delta t\right)+O\left(\Delta t^{1.5}\right)$$

Here, $\Delta {\vec{W}}_{t}$ is a random sample from $N^{d}\left(0,\Delta t\right)$. Note that the Milstein scheme is only accurate up to O(Δt), and this error can accumulate through the course of the simulation. Therefore, to control the error in the numerical update scheme, we will adaptively choose the time step Δt for updates by setting a local error tolerance. Local error at time *t* is defined as the deviation between a single step update made with time step Δt and a two-step update made with the time step Δt/2 each.
(9)$$err\left({\vec{x}}_{t},\Delta t\right):=\frac{∥\left(M\left({\vec{x}}_{t},\Delta {\vec{W}}_{1},\frac{\Delta t}{2}\right)+M\left({\vec{x}}_{t+\frac{\Delta t}{2}},\Delta {\vec{W}}_{2},\frac{\Delta t}{2}\right)\right)-M\left({\vec{x}}_{t},\Delta {\vec{W}}_{t},\Delta t\right)∥}{\sqrt{d}}$$

Here, we omitted the dependence of M on $\vec{\mu }$ and σ for brevity. $\Delta {\vec{W}}_{1}$ and $\Delta {\vec{W}}_{2}$ are chosen in such a way that $\Delta {\vec{W}}_{1}+\Delta {\vec{W}}_{2}=\Delta {\vec{W}}_{t}$. To satisfy the local error tolerance Tol, we need to choose Δt, such that $err({\vec{x}}_{t},\Delta t)<Tol$. The exact prescription on how this choice is made can be found here [40].

Computing Γ requires computing the derivative of $P({\vec{x}}_{t},t)$ of the numerical solutions. Since derivatives are sensitive to numerical noise, we need accurate estimates of the probability distribution. This is achieved by numerically solving a large number of SDEs (we use at least $2\times 10^{7}$ samples in this work). This is an impractically large number for most computers. Numerically finding $2\times 10^{7}$ solutions, with each requiring around 10,000 time steps (depending on the required accuracy) will require at least 1600 GB of memory, assuming 64-bit floating point values. Additionally, assuming a fast 2 ms per solution, the entire simulation will take around 11 h if carried out serially. To solve these problems, we use GPU computing. With GPU computing [41], we can perform updates on a set of values $\left\{{\vec{x}}_{t}\right\}$, simultaneously using Equation (Equation 7) to get $\left\{{\vec{x}}_{t+\Delta t}\right\}$. $\left\{{\vec{x}}_{t}\right\}$ can then be removed from memory after computing the desired statistics, making it memory efficient. As for the runtime, a GPU-based parallel implementation [42] in Python takes around 4 min to simulate $2\times 10^{7}$ samples for 10,000 time steps on a consumer laptop equipped with Nvidia RTX 2080 GPU. See Appendix A for detailed scaling relations of simulation runtime.

### 2.2. Estimating Γ

The form of Γ in Equation (Equation 4) makes it unsuitable for numerical computation due to the presence of p(x,t) term in the denominator, which can become zero. We therefore rewrite the equation using the redefinition $q\left(x,t\right):=\sqrt{p(x,t)}$.
(10)$$\Gamma^{2}\left(t\right)=4\int dx{\left[\frac{\partial q(x,t)}{\partial t}\right]}^{2}$$

To calculate Γ, we first estimate the PDF using histograms. (It would be more accurate to use kernel density estimators [43], but that is more computationally expensive). The derivative in Equation (Equation 10) is approximated as a finite difference and the integral as a Riemann sum. The specific methods and the error estimates are provided in Appendix B. After computing Γ, information length can be computed from the numerical integration of Equation (Equation 3).

A major source of error in the estimation of Γ is due to the compact support and overlap of numerically estimated probability densities. (This will also be a source of error with numerical FPE solvers). During the simulation, time steps are chosen adaptively by setting a tolerance for the local numerical error. However, in some regimes, this results in the system evolving too fast, so that there is little or no overlap between the probability densities at adjacent time steps (Figure 1). Theoretically, these densities (Figure 1) have the support of the entire real line, and the only error will be caused by approximating the derivative and the integral. However, in practice, we are running the MC simulation with a finite sample size, and computers have finite numerical precision; as such, the estimated densities have compact support. Figure 2 shows how the amount of overlap between the densities affects the error in Γ estimate. There are two main sources of sub-optimal overlap: changes in mean and changes in standard deviation.

Consider Δt chosen adaptively to satisfy the local error tolerance. Now assume for some $\tilde{\Delta t}$, we get optimal overlap. If $\tilde{\Delta t}>\Delta t$, we can wait for a few steps before estimating the Γ. However, if $\tilde{\Delta t}<\Delta t$, we can choose a $\tilde{\Delta t}$ as a temporary time step and generate a collection of points $\left\{{\vec{x}}_{t+\tilde{\Delta t}}\right\}$ from $\left\{{\vec{x}}_{t}\right\}$ using Equation (Equation 7), which results in optimal overlap between densities. The local error will still be smaller than the tolerance since $\Delta t<\tilde{\Delta t}$. In order to derive the value of $\tilde{\Delta t}$, we use the Milstein update scheme and restrict ourselves to the one-dimensional case for simplicity. First, we look at the change in mean value.
(11)$$x_{t+\Delta t}=x_{t}+\mu \left(x_{t}\right)\Delta t+\sigma \left(x_{t}\right)\left(x_{t}\right)\Delta W_{t}+\frac{1}{2}\sigma \left(x_{t}\right)\sigma^{\prime }\left(x_{t}\right)\left(\left(\Delta W_{t}^{2}\right)-\Delta t\right)+O\left(\Delta t^{1.5}\right)$$

Taking the expectation value of both sides, we get:(12)$$E\left[x_{t+\Delta t}\right]=E\left[x_{t}\right]+E\left[\mu (x_{t})\right]\Delta t+O\left(\Delta t^{1.5}\right)$$

Here, we use the fact that $E\left[\Delta W_{t}\right]=0$ and $E\left[{\left(\Delta W_{t}\right)}^{2}\right]=\Delta t$. Note that $x_{t}$ and $\Delta W_{t}$ are independent random variables. For *X* and *Y*, the independent random variables are $E\left[XY\right]=E\left[X\right]E\left[Y\right]$.

In order to achieve optimal overlap (Figure 2 (left)), we need $E\left[x_{t+\Delta t}\right]-E\left[x_{t}\right]=0.2\mathrm{Std}\left[x_{t}\right]$.
(13)$${\tilde{\Delta t}}_{mean}=\frac{0.2\mathrm{Std}\left[x_{t}\right]}{E\left[\mu (x_{t})\right]}$$

Now to derive the effect on change in standard deviation on $\tilde{\Delta t}$, a similar calculation for $\mathrm{Var}\left[x_{t+\Delta t}\right]$ can be performed, which yields:(14)$$\mathrm{Var}\left[x_{t+\Delta t}\right]=\mathrm{Var}\left[x_{t}\right]+\left(2\mathrm{Cov}\left[\mu \left(x_{t}\right),x_{t}\right]+E\left[\sigma^{2}\left(x_{t}\right)\right]\right)\Delta t+O\left(\Delta t^{1.5}\right)$$

Now, in order to achieve optimal overlap, we need $\mathrm{Var}\left[x_{t+\Delta t}\right]/\mathrm{Var}\left[x_{t}\right]=0.9^{\pm 2}$ (Figure 2 (right)). $0.9^{+2}$ when $\mathrm{Var}\left[x_{t+\Delta t}\right]<\mathrm{Var}\left[x_{t}\right]$ and $0.9^{-2}$ when $\mathrm{Var}\left[x_{t+\Delta t}\right]>\mathrm{Var}\left[x_{t}\right]$.
(15)$${\tilde{\Delta t}}_{std}=\mathrm{Var}\left[x_{t}\right]\left|\frac{0.9^{2}-1}{2\mathrm{Cov}\left[\mu \left(x_{t}\right),x_{t}\right]+E\left[\sigma^{2}\left(x_{t}\right)\right]}\right|$$

Note that we have only considered the first and second moments here. It is potentially possible to improve the accuracy of the Γ estimate by considering higher moments. However, this improvement will be marginal, since overlap between two distributions is most affected by its first and second moments.

After calculating both the time steps ${\tilde{\Delta t}}_{mean}$ and ${\tilde{\Delta t}}_{std}$, we take the minimum of the two to perform the update on $\left\{x_{t}\right\}$ to get $\left\{x_{t+\tilde{\Delta t}}\right\}$, where $\tilde{\Delta t}=\min\left({\tilde{\Delta t}}_{mean},{\tilde{\Delta t}}_{std}\right)$. After the estimation, the Γ$\left\{x_{t+\tilde{\Delta t}}\right\}$ is discarded and we continue the simulation with $\left\{x_{t+\Delta t}\right\}$. This prevents any significant slowing in the simulation if $\tilde{\Delta t}\ll \Delta t$. Note that calculating ${\tilde{\Delta t}}_{mean}$ and ${\tilde{\Delta t}}_{std}$ using the entire set of points $\left\{x_{t}\right\}$ will be computationally expensive. Only a small subset is used to perform this calculation.

After Γ is estimated, we can calculate the information length by approximating the integral as a Riemann sum.

Looking at Figure 3, it can be seen than the percentage error in Γ blows up towards the end of the simulation. This is when the system approaches a stationary state and the probability density stops evolving. The exact Γ reaches zero, whereas the numerical estimate will have a small nonzero error (Figure 4). Even though the percentage error in Γ becomes large, the absolute error remains small (Figure 2 (left)) and will have minimal contribution to the error in Information Length calculation. However, when the initial distribution is ’closer’ to the stationary distribution (small $x_{0}$ values in Figure 3), the absolute value of Information Length will be small, and the error in Γ estimate will have a more significant contribution to the Information Length calculation.

It is to be noted that measuring Γ of multi-dimensional problems is a significant computational challenge which requires further investigation. MC SDE simulation is better at handling such problems compared with FPE solvers due to its linear scaling with number of dimensions. However, this comes at the price of accuracy when estimating PDFs of higher dimensional problems. It is still possible to accurately study Γ from marginal PDFs of multi-dimensional problems using MC SDE simulation.

A python implementation of SDE Simulation, along with the Γ measurement used in this work, can be found here [42].

## 3. Linear vs. Cubic Statistics

In this section, we will use methods developed in the previous section and study nonlinear damping of the Information Geometry of a stochastic process. We will compare the Ornstein–Uhlenbeck process [44], a model for prototypical linear driven dissipative process, defined by linear SDE:(16)$$dx_{t}=-\theta x_{t}dt+\sigma dW_{t}$$
with a cubic SDE defined by the equation:(17)$$dx_{t}=-\theta x_{t}^{3}dt+\sigma dW_{t}$$

The cubic damping term has been previously used to model ${\mathrm{CO}}_{2}$ emissions [45], phase transitions [24], and self-organized shear flows [46].

Throughout this section, unless otherwise stated, we restrict ourselves to the values θ=1.0 and σ=0.1. The initial distribution is always $N\left(x_{0},10^{-10}\right)$, and we run the SDE simulation from t=0 to t=100. The data in this section were generated using $2\times 10^{7}$ samples, and PDF was approximated by a histogram with 703 bins, which was chosen using an empirical formula $2.59\sqrt[{3}]{n}$. The time steps were adaptively chosen by setting local error tolerance at $5\times 10^{-4}$.

Two of the simplest statistics that can be measured are the mean and the standard deviation of the distribution (Figure 5 and Figure 6). The trends in mean can be readily seen by taking expectation value on both sides of Equations (Equation 16) and (Equation 17). For the linear SDE, we have $dE\left[x_{t}\right]=-\theta E\left[x_{t}\right]dt$, which can be solved to get $E\left[x_{t}\right]=x_{0}e^{-t}$. For the Cubic SDE, we can follow similar steps:(18)$$\begin{array}{cc} dE\left[x_{t}\right] & =-\theta E\left[x_{t}^{3}\right]dt \end{array}$$(19)$$\begin{array}{cc} dE\left[x_{t}\right] & \approx -\theta E{\left[x_{t}\right]}^{3}dt \end{array}$$(20)$$\begin{array}{cc} \Rightarrow E\left[x_{t}\right] & \approx \frac{1}{\sqrt{\frac{1}{x_{0}^{2}}+2\theta t}} \end{array}$$

The approximate solution of the mean value of Cubic SDE is only valid when the standard deviation of the distribution is much smaller than the mean (Appendix C). Note that in Equation (20), when $1/x_{0}^{2}\ll \theta t$, $E\left[x_{t}\right]\approx {\left(2\theta t\right)}^{-0.5}$, making the trajectory independent of $x_{0}$. This can be seen in Figure 5 (right), when all the lines merge into one. However, around t=5, the approximation fails, since the value of the standard deviation (Figure 6 (right)) and the mean (Figure 5 (right)) becomes comparable. Therefore, the curves fail to follow the same trajectory $E\left[x_{t}\right]\approx {\left(2\theta t\right)}^{-0.5}$ for t>5.

The trends in standard deviation of the Linear SDE are readily explained by Equation (Equation 39), which does not depend on initial mean position $x_{0}$, but only on the initial standard deviation, the drift coefficient, and the diffusion coefficient. As we write this paper, no exact analytic solution exists for the Cubic SDE. However, an approximate analytic treatment can be found here [47].

We define the asymptotic Information Length ($L_{\infty }$) to be the Information Length it took for the system to reach the stationary state of its PDF.
(21)$$L_{\infty }:=L\left(t\to \infty \right)$$

Analytically, a SDE reaches its stationary state as t→∞. However, numerically, we see that the probability density stops evolving after a finite time. We can see this from Figure 7, as L becomes a constant. L will still continue to increase slightly due to numerical error, but this contribution will be negligible, as shown in Figure 3. $L_{\infty }$ measures how many statistically distinguishable states the system passes through to reach its stationary state. From Figure 6, it is evident that, compared to the linear SDE, the PDF of the cubic SDE undergoes a lot more change before reaching its stationary state, for larger values of $x_{0}$. For small values of $x_{0}$, the trend in standard deviation is similar between linear and cubic SDEs, since the initial evolution of the cubic process is Gaussian. This is confirmed in Figure 7, which shows that for large values of $x_{0}$, $L_{\infty }$ is significantly larger for the cubic SDE for same values of $x_{0}$ and, for small $x_{0}$ values, the $L_{\infty }$ values are comparable.

### Information Length Scaling

In Figure 7, we have already seen that $L_{\infty }$ depends on $x_{0}$ and has different behavior for linear and cubic SDEs. In [47], by numerically solving the FPE (also analytically for the linear SDE), it was shown that for large values of $x_{0}$, $L_{\infty }$ shows different scaling behavior for linear and cubic SDEs. For the linear SDE, $L_{\infty }∼x_{0}$. For the cubic SDE, $L_{\infty }∼x_{0}^{m}$, where 1.5<m<1.9. We reproduce this result in Figure 8. Note that, since $L_{\infty }$ is a dimensionless quantity, it is not possible to derive these values theoretically using dimensional analysis. In the absence of general analytic tools to study this property, numerical methods are indispensable.

## 4. Unequal Time Joint PDF: Bimodality for Cubic Force

A clear advantage of MC SDE Simulation over solving FPE is the ability to compute the unequal time joint PDFs $P\left(x\left(t_{1}\right),x\left(t_{2}\right)\right)$. Unequal time joint PDFs have been previously used for causality and to establish causal relations. In this section, we showcase the ability of MC SDE simulation to estimate $P\left(x\left(t_{1}\right),x\left(t_{2}\right)\right)$ and study its properties.

### 4.1. Unequal Time PDFs in the Stationary State

For the linear SDE $P\left(x\left(t_{1}\right),x\left(t_{2}\right)\right)$ is always a Gaussian, but with a covariance matrix that depends on $t_{1}$ and $t_{2}$. However, for the cubic SDE, we see the emergence of a bimodal distribution (Figure 9 (right)) depending on $t_{1}$ and $t_{2}$ values. This behavior prevails even after the system has reached its stationary state, but now only depends on the difference $\Delta t:=t_{2}-t_{1}$. The bimodality indicates a finite memory induced by the non-linearity. In order to further study this behavior, we first ensure the system has reached a stationary state by simulating from t=0 to t=1500 with a sample size of $8\times 10^{7}$ for the cubic process. After setting $t_{1}=1500$, $t_{2}$ values are chosen from the data generated by further simulating the system for 300 time units. The PDF was approximated by a histogram with 60×60 bins. The time steps were adaptively chosen by setting the local error tolerance as 0.01σ.

Note that, because of the symmetry of diffusion term and the drift term, the bimodality of $P\left(x\left(t_{1}\right),x\left(t_{2}\right)\right)$ for cubic SDE is always symmetric with respect to the line $x\left(t_{1}\right)=x\left(t_{2}\right)$. Therefore to make the numerical study of the bimodality easier, we restrict our attention to the diagonal of the joint probability density $P\left(x\left(t_{1}\right)=x\left(t_{2}\right)\right)$, which is then normalized to integrate to one.

To quantify the bimodality of $P\left(x\left(t_{1}\right)=x\left(t_{2}\right)\right)$, we use the following fitting function to perform a nonlinear fit and estimate the parameters.
(22)$$P\left(x\left(t_{1}\right)=x\left(t_{2}\right)\right)=a\left[\exp\left(-{\left(\frac{x}{c}\right)}^{4}\right)+b\exp\left(-{\left(\frac{x-e}{d}\right)}^{2}\right)+b\exp\left(-{\left(\frac{x+e}{d}\right)}^{2}\right)\right]$$

To motivate this fitting function, note that for Δt=0, we get a purely quartic exponential, since it is nothing but the stationary state $P_{s}\left(x\right)$ of the cubic SDE. For Δt→∞, the correlation between points reaches zero, $P\left(x\left(t_{1}\right),x\left(t_{2}\right)\right)∼P_{s}\left(x_{1}\left(t_{1}\right)\right)P_{s}\left(x_{2}\left(t_{2}\right)\right)$, the product of two independent stationary distributions. The quadratic exponential terms are motivated by Gaussian distribution of the noise, and are found to describe the data accurately. There are two quadratic exponential terms, since the SDE is symmetric about the point x=0 and the bimodal peaks occur symmetrically on opposite sides.

In the fitting function, Parameter *a* is a measure of the overall height of the density curve. Parameter *b* represents the ratio of contribution from the quadratic exponential to the quartic exponential, and denotes the degree of bimodality. Parameter *e* is the location of the peaks. Parameter *c* is a measure of the overall spread of the density function, while parameter *d* is a measure of the spread of bimodal peaks. Note that a nonzero value for parameter *e* and *b* denotes bimodality in the distribution.

From Figure 10, we can see that the joint PDF becomes bimodal for a range of Δt values, since parameters *b* and *e* have nonzero values. The standard deviation of the quadratic term denoted by parameter *d* is almost a constant for a fixed σ, whereas the location of the bimodal peak denoted by parameter *e* changes slightly, but never becomes zero. That means that while transitioning from a bimodal to a unimodal distribution, the bimodal peaks do not continuously merge into one another, but slowly become less prominent and eventually disappear, as inferred from the value of parameter *b*. The artifacts towards the end of the curves of parameters *b* and *e* are due to the fact that it is not possible to consistently fit parameters *b* and *e* when there is negligible contribution from the quadratic exponential term as b→0.

The values of parameter *a* and *c* in Figure 10 can be understood by first considering the Δt→0 and Δt→∞ limits. For Δt→0, we have $P\left(x\left(t_{1}\right),x\left(t_{2}\right)\right)=\delta \left(x\left(t_{1}\right)-x\left(t_{2}\right)\right)P_{s}\left(x\right)$, where $P_{s}\left(x\right)$ is the stationary state. The diagonal of joint PDF then becomes $P\left(x\left(t_{1}\right)=x\left(t_{2}\right)\right)=P_{s}\left(x\right):=a_{s}\exp\left(-{\left(x/c_{s}\right)}^{4}\right)$. Here, after integrating *x*, we can find, $a_{s}\approx 0.55/c_{s}$ for all values of σ, which can be numerically verified. When Δt→∞, since there is no correlation, $P\left(x\left(t_{1}\right),x\left(t_{2}\right)\right)=P_{s}\left(x\left(t_{1}\right)\right)P_{s}\left(x\left(t_{2}\right)\right)$. The normalized diagonal then becomes $P\left(x\left(t_{1}\right)=x\left(t_{2}\right)\right)∼\exp\left(-2{\left(x/c_{s}\right)}^{4}\right)$. The normalization factor can be derived by integrating out *x*. Therefore, when Δt→∞, we have $a=2^{1/4}a_{s}$ and $c=c_{s}/2^{1/4}$, which agrees with numerical results. For intermediate Δt values, there will be correlation, and the behavior cannot be easily explained.

The trends in Figure 10 for different σ values can be explained by looking at scaling relations. For the cubic equation, we have $dx_{t}=-\theta x_{t}^{3}dt+\sigma dW_{t}$. Looking at the individual terms, we can infer the dimensions, $\theta ∼x^{-2}t^{-1}$ and $\sigma ∼x^{1}t^{-1/2}$. Therefore, we expect the following scaling relations:(23)$$\begin{array}{cc} t & ∼\frac{1}{\theta^{1/2}\sigma } \end{array}$$(24)$$\begin{array}{cc} x & ∼\theta^{-1/2}\sigma^{1/2} \end{array}$$

These relations are numerically verified in Figure 11 for a fixed value of θ, for parameters *a*, *c*, *d*, *e*, and the relationship between noise and Δt, corresponding to peak value of parameter *b*. Note that $a∼\sigma^{0.5}$ in Figure 11, because parameter *a* is a normalization constant. $\int ap\left(x\right)dx=1\Rightarrow a∼x^{-1}∼\theta^{1/2}\sigma^{-1/2}$. The peak value of parameter *b* seems to be a constant with the value of 1/3. It cannot be derived from simple scaling arguments alone; further investigation is required to understand its origin.

### 4.2. Evolution of Bimodality in the Non-Stationary State

In this section, we will look at how the bimodality in the joint PDF evolves qualitatively before p(x,t) has reached a stationary state. Since bimodality occurs only for the cubic process, the following results are for the SDE $dx_{t}=-1.0x_{t}^{3}dt+0.1dW_{t}$. To this end, we follow the same prescription from Section 4, but with a slightly modified version of the fitting function (Equation (Equation 22)).
(25)$$P\left(x\left(t_{1}\right)=x\left(t_{2}\right)\right)=a\exp\left(-{\left(\frac{x}{c}\right)}^{4}\right)+b\exp\left(-{\left(\frac{x-e}{d}\right)}^{2}\right)+b\exp\left(-{\left(\frac{x+e}{d}\right)}^{2}\right)$$

This modification is made since for some $t_{1}$ and $t_{2}$ values, there is no contribution from the quartic exponential term, unlike in the case of stationary state, where there is always a quartic exponential contribution to the distribution.

The nonstationary phase exhibits (Figure 12) rich behavior, which asymptotically transitions to the stationary state behavior as $t_{1}$ becomes large. For small $t_{1}$ values, Parameter *a* ≪ Parameter *b*, since $P\left(x\left(t_{1}\right),x\left(t_{2}\right)\right)$ has very little contribution from the quartic exponential term. This is because the initial distribution is a Gaussian distribution, and it is slowly evolving towards the quartic exponential distribution in the stationary state. For larger $t_{1}$ values, Parameter *a* starts dominating, since $P\left(x\left(t_{1}\right),x\left(t_{2}\right)\right)$ has predominant contribution from quartic exponential term, as expected of the system reaching its quartic exponential stationary state. For some intermediate $t_{1}$ values, depending on Δt values, we see an interesting behavior where $P\left(x\left(t_{1}\right),x\left(t_{2}\right)\right)$ becomes purely a quartic exponential (Parameter *b* = 0) twice before becoming a mixture of quadratic and quartic exponential terms. Further investigation into this behavior is not undertaken at this time, and only serves to demonstrate the potential capabilities of GPU-accelerated MC SDE simulation for future work.

## 5. Discussion

In this work, we developed a method for fast and accurate study of the Information Geometry of SDEs using Monte Carlo simulation. We identified the computational challenges and overcame them by using GPU computing. Specifically, the major limitation with MC SDE simulation in the estimation of Γ was the sub-optimal overlap of PDFs at subsequent time points. We solved this problem by developing an interpolation method to compute PDFs with optimal overlap.

As an application of the new method, we compared the Information Geometry of SDEs with a linear and a cubic damping force. We were able to reproduce the analytic results for the linear SDE and the previous numerical results [34] for cubic SDE obtained using FPE solvers. This was particularly true of large values of $x_{0}$, $L_{\infty }∼x_{0}$ for the linear case and $L_{\infty }∼x_{0}^{m}$ for the cubic case, where m=1.88 when θ=1.0 and σ=0.1. We further showcased the advantage of MC SDE simulation over FPE solver by computing the joint PDF $P\left(x\left(t_{1}\right),x\left(t_{2}\right)\right)$. Unlike the linear SDE, the cubic SDE led to an interesting bimodal PDF $P\left(x\left(t_{1}\right),x\left(t_{2}\right)\right)$, which was observed even after reaching a stationary state. After reaching the stationary state $P\left(x\left(t_{1}\right),x\left(t_{2}\right)\right)$ only depends on $\Delta t=t_{2}-t_{1}$. In the stationary state, we further studied the bimodality by quantifying it and looking at the power-law scaling relations with respect to noise levels σ and provided theoretical scaling arguments. The maximum value of the ratio of quadratic to quartic contribution (Parameter *b*) is found to be a constant 1/3 irrespective of the noise levels, which requires further study. Finally, we qualitatively looked at $P\left(x\left(t_{1}\right),x\left(t_{2}\right)\right)$ in the nonstationary state. The MC SDE simulation can be an important tool for further studying this behavior.

It is important to note that the methods that we developed here for one stochastic variable are general, and can be extended for more than one variable, as well as for investigating different metrics or thermodynamic quantities. These will be addressed in future work. Furthermore, it will be of interest to investigate fast implementations of Kernel Density Estimators [48,49] which will improve the accuracy of joint PDF estimates. We note that in numerical experiments, it was seen that compared with histograms, using Kernel Density Estimators to estimate PDFs provided 2–5 times reduction in error with identical PDFs, albeit with a performance trade-off.

## Figures and Tables

**Figure 1 entropy-24-01113-f001:**
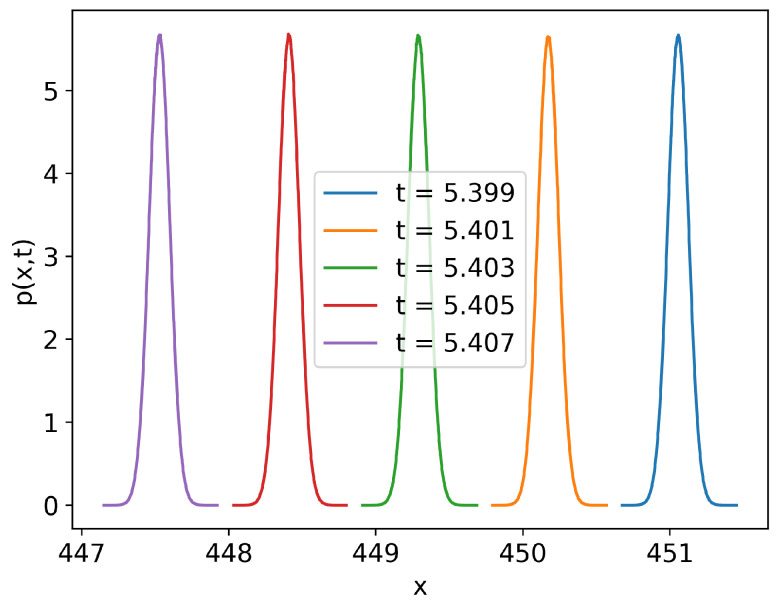
Probability density at adjacent time steps for the SDE $dx_{t}=-x_{t}dt+0.1dW_{t}$ with initial normal distribution $N(10^{5},10^{-10})$. Time steps were chosen adaptively to limit local error to $5\times 10^{-4}$. The specific time interval was chosen to showcase the lack of overlap between PDFs.

**Figure 2 entropy-24-01113-f002:**
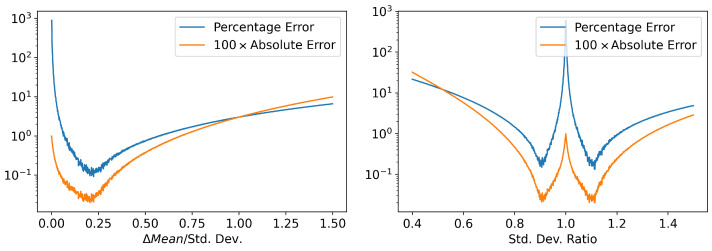
(**left**) Error in Γ calculation for two Gaussians with same standard deviation (Std. Dev. = 1) but with different means. The error estimate of Γ is lowest when ΔMean≈0.2Std.Dev. (**right**) Error in Γ calculation for two Gaussians with same Mean (Mean = 0) but with different standard deviation. The error Γ estimate is lowest when ratio of standard deviation is approximately 0.9 or 1.1 (≈0.9${}^{-1}$). Discretized version of Equation (Equation 10) was used for the estimate and PDF was approximated by a histogram with 703 bins. We considered $2\times 10^{7}$ samples for each distribution. dt is chosen to be 1. Estimates were repeated 40 times and mean of the error was taken.

**Figure 3 entropy-24-01113-f003:**
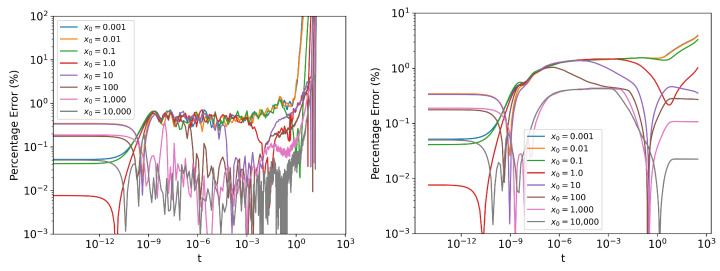
Error in estimated (**left**) Γ and (**right**) L by simulating $2\times 10^{7}$ parallel instances of the Equation $dx_{t}=-1.0x_{t}dt+0.1dW_{t}$ with initial values sampled from the normal distribution $N(x_{0},10^{-10})$ and computing the PDFs using histograms with 703 bins. The local error tolerance was $5\times 10^{-4}$. Note that an initial step size $\Delta t=10^{-14}$ was chosen to produce more frequent estimates in the initial part of the simulation. See Appendix D for exact solution.

**Figure 4 entropy-24-01113-f004:**
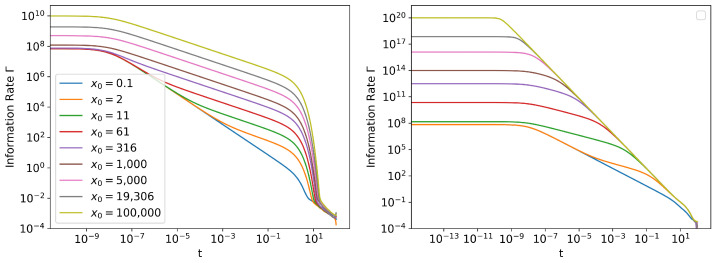
The Γ of (**left**) Linear SDE and (**right**) Cubic SDE for different initial conditions $N(x_{0},10^{-10})$, as described in Section 3. Note that towards the end of the range of *t*, the values will be dominated by errors, as shown in Figure 3. Therefore instead of dropping to zero, they will have a finite nonzero value. For exact solution of Linear SDE, see Appendix D.

**Figure 5 entropy-24-01113-f005:**
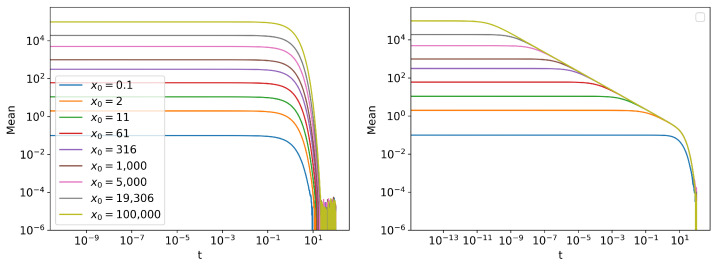
The mean of the distribution for different initial conditions $N(x_{0},10^{-10})$ of (**left**) linear SDE and (**right**) cubic SDE.

**Figure 6 entropy-24-01113-f006:**
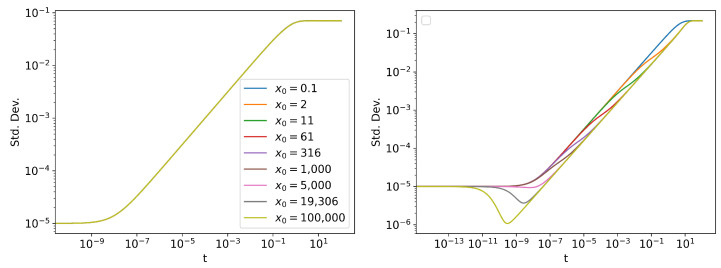
The standard deviation of the distribution for different initial conditions $N(x_{0},10^{-10})$ of (**left**) linear SDE and (**right**) cubic SDE. Note that for the linear SDE, all the lines overlap.

**Figure 7 entropy-24-01113-f007:**
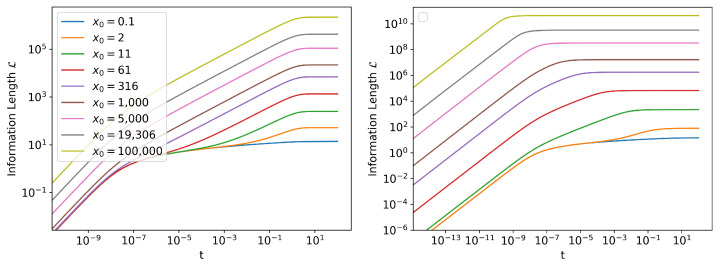
The Information Length of (**left**) linear SDE and (**right**) cubic SDE for different initial conditions $N(x_{0},10^{-10})$. For exact solution of linear SDE, see Appendix D.

**Figure 8 entropy-24-01113-f008:**
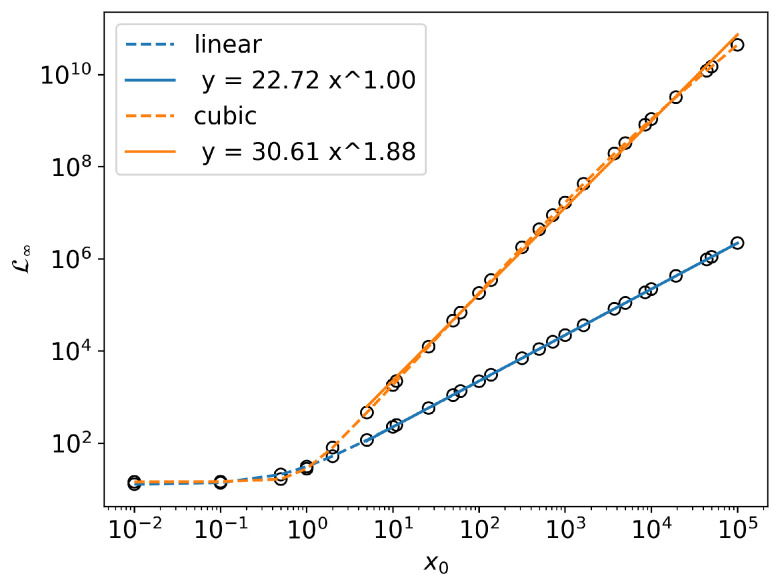
Scaling behavior of $L_{\infty }$ with respect to $x_{0}$ for linear and cubic SDE with θ=1.0 and σ=0.1.

**Figure 9 entropy-24-01113-f009:**
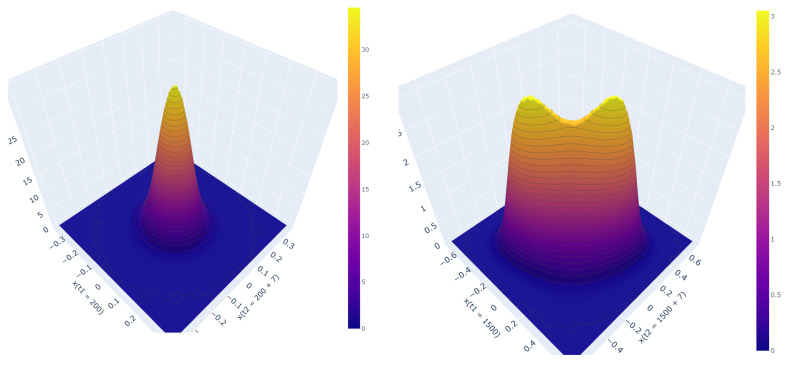
$P\left(x\left(t_{1}\right),x\left(t_{2}\right)\right)$ for (**left**) linear SDE with $t_{1}=200$ and $t_{2}=207$ and (**right**) cubic SDE with $t_{1}=1500$ and $t_{2}=1507$. For both the SDEs, γ=1.0 and σ=0.1. Both SDEs have reached their stationary states.

**Figure 10 entropy-24-01113-f010:**
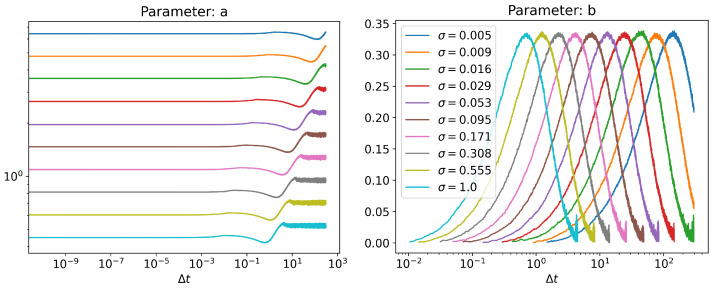
Panels show the values of *a*, *b*, *c*, *d* and *e* parameters of Equation (Equation 22), obtained through nonlinear function fitting at each time point $t_{2}$, expressed as a function of $\Delta t:=t_{2}-t_{1}$ for different noise levels σ. To ensure the density function p(x,t) for the cubic process reached a stationary state, we chose $t_{1}=1500$. Note that for parameters *b*, *d* and *e*, the domain of the plots is restricted to the region where parameter *b* is nonzero. Noise starts dominating outside this region, since there is negligible contribution from the quadratic terms, and nonlinear fit cannot find a consistent unique value for these parameters.

**Figure 11 entropy-24-01113-f011:**
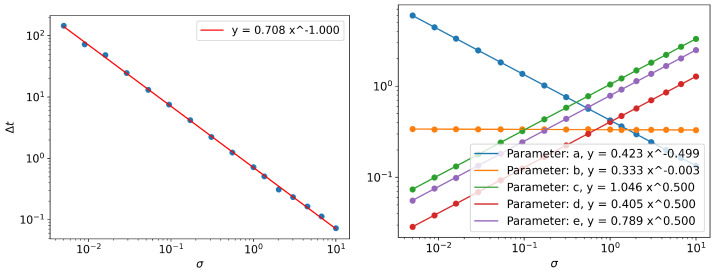
(**left**) Scaling behavior of peak poisition of parameter *b* with respect to noise level σ and (**right**) Scaling behavior of parameter values corresponding to the peak position of parameter *b*.

**Figure 12 entropy-24-01113-f012:**
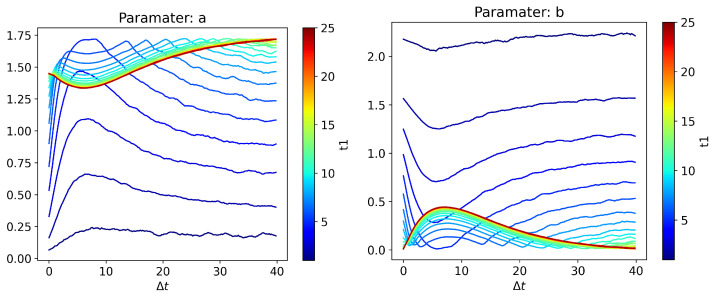
Behavior of (**left**) Parameter *a* and (**right**) Parameter *b* for different $t_{1}$ and $t_{2}=t_{1}+\Delta t$ values.

**Table 1 entropy-24-01113-t001:** Comparison between grid-based FPE solver and MC SDE simulation.

	Grid-Based FPE Solver	MC SDE Simulation
Accuracy	Depends on grid size. No well-defined prescription on choosing grid-size.	Depends on number of samples(*n*) [36]. Typically less accurate for practical sample sizes.
Boundary condition	Requires carefully chosen non-trivial boundary conditions. Cannot handle discontinuous initial conditions such as Dirac delta function.	Requires only an initial distribution as boundary condition.
Memory Usage & Runtime	Scales exponentially with dimension *d*. $O\left(n_{1}n_{2}\ldots n_{d}\right)$. Here, $n_{1},n_{2},\ldots ,n_{d}$ are the number of grid points along each dimension.	Scales linearly with dimension *d*. $O\left(nd\right)$. Here, *n* is the number of samples.
Correlation Study	Cannot study correlations and associated memory effects using FPE.	Can study correlations. See Section 4 for unequal time joint PDF estimates.

## Data Availability

The data that support the findings of this study are available from the corresponding author, A.A.T., upon reasonable request.

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
