# Peer review of "Monte Carlo Simulation of Stochastic Differential Equation to Study Information Geometry"

_entropy, 2022, doi:10.3390/e24081113_

Round 1

Reviewer 1 Report

This paper studies Monte Carlo simulations of stochastic differential equations and related information geometric measurements. The contents are overall interesting and worthy to be published. I have the following comments for the authors to work on a revision.

Fig 1. Why the initial distribution is not presented?

Fig 3. Are these SDE simulations based on the average over multiple runs?

eq.(8) here, the expression of the information rate should be instantiated based on the parametric form of q(x,t). In all the experiments, please give the closed-form of information rate if they are available.

Explain in section 1 how the information length is computed based on the information rate (e.g. based on numerical integration).

Section 4. Why in this section the information rate and length are not evaluated? (all experiments are on the behavior of the parameters of the PDFs). Please include related experimental results on how these two intrinsic quantities evolve with time. This may require further experiments.

Note to compute the information length, one can use the KL divergence as an approximation of the small distances. This can be mentioned if not used in the experiments.

after eq(5), explain the notation \mathcal{M}

Reviewer 2 Report

In this paper, the authors discuss the possibility of using the Monte Carlo method to evaluate the time evolution of stochastic differential equations. The method utilizes the Milstein scheme and compared its performance against numerically solving the Fokker-Planck equation.

 The idea is straight forward and the results are fair. However, some terminology used in the paper is not acceptable and the authors must rewrite those parts, otherwise, this paper cannot be accepted for publication.

This reviewer does not understand what the authors mean by “Information Geometry.” Information geometry is a field to study the set of probability with the method of differential geometry. The authors do not cite any literature on information geometry and the paper has nothing to do with it. They just defined (1) and (2) as information geometry, which is totally misleading. Estimating (2) is fine, but do not call this information geometry.

The authors also use the term “Information Rate” which is again misleading. Information rate is a basic term used for a different quantity in information theory. 

When some terms have already been widely used for an important concept, you should pay respect to that. You should also cite appropriate references for each term, not only those of the authors.

Round 2

Reviewer 2 Report

okay, you  can accept this version.